# *pilE* G-Quadruplex Is Recognized and Preferentially Bound but Not Processed by the MutL Endonuclease from *Neisseria gonorrhoeae* Mismatch Repair Pathway

**DOI:** 10.3390/ijms24076167

**Published:** 2023-03-24

**Authors:** Viktoriia Yu. Savitskaya, Vadim V. Strekalovskikh, Viktoriia G. Snyga, Mayya V. Monakhova, Alexander M. Arutyunyan, Nina G. Dolinnaya, Elena A. Kubareva

**Affiliations:** 1Department of Chemistry, Lomonosov Moscow State University, 119991 Moscow, Russia; 2Department of Bioengineering and Bioinformatics, Lomonosov Moscow State University, 119991 Moscow, Russia; 3Belozersky Institute of Physico-Chemical Biology, Lomonosov Moscow State University, 119991 Moscow, Russia

**Keywords:** G-quadruplex, DNA mismatch repair, MutS, MutL, pilin, *Neisseria gonorrhoeae*, antigenic variation

## Abstract

The human pathogen *Neisseria gonorrhoeae* uses a homologous recombination to undergo antigenic variation and avoid an immune response. The surface protein pilin (PilE) is one of the targets for antigenic variation that can be regulated by *N. gonorrhoeae* mismatch repair (MMR) and a G-quadruplex (G4) located upstream of the *pilE* promoter. Using bioinformatics tools, we found a correlation between *pilE* variability and deletion of DNA regions encoding ngMutS or ngMutL proteins, the main participants in *N. gonorrhoeae* methyl-independent MMR. To understand whether the G4 structure could affect the ngMutL-mediated regulation of pilin antigenic variation, we designed several synthetic *pilE* G4-containing oligonucleotides, differing in length, and related DNA duplexes. Using CD measurements and biochemical approaches, we have showed that (i) ngMutL preferentially binds to *pilE* G4 compared to DNA duplex, although the latter is a cognate substrate for ngMutL endonuclease, (ii) protein binding affinity decreases with shortening of quadruplex-containing and duplex ligands, (iii) the G4 structure inhibits ngMutL-induced DNA nicking and modulates cleavage positions; the enzyme does not cleave DNA within G4, but is able to bypass this noncanonical structure. Thus, *pilE* G4 may regulate the efficiency of pilin antigenic variation by quadruplex binding to ngMutL and suppression of homologous recombination.

## 1. Introduction

The host-adapted human pathogen *Neisseria gonorrhoeae* (also known as gonococcus) is the causative agent of the sexually transmitted disease gonorrhea [1]. The ongoing incidence of gonorrhea worldwide, coupled with growing antimicrobial resistance and using various approaches to avoid recognition by the human immune system [2], highlights the need to better understand the molecular basis of *N. gonorrhoeae* infection.

Drug resistance is associated with antigenic and phase variation in *N. gonorrhoeae* surface-exposed structures that are used to attach to host cells and evade the immune system [3]. These surface structures include type IV pili, lipooligosaccharide (LOS), porin (PorAB), and opacity (Opa) proteins. Type IV pili variation is an important virulence factor required for complete infectivity [4,5]; it mediates several steps in *N. gonorrhoeae* pathogenesis, including the first, adherence to human epithelial cells [6].

The process of phase variation generally requires the presence of repetitive genomic sequences within or near coding regions. The instability of these repeats during replication can shift the reading frames or change the strength of promoters [7], thereby affecting the expression of *N. gonorrhoeae* surface components. Antigenic variation arises through changes in the amino acid sequence of the pilin (PilE) monomer, the predominant component of type IV pili.

The genome of gonococcal strain FA1090 contains one complete pilin expression locus, *pilE*, which consists of conservative and variable regions, and highly homologous *pilS* loci without promoters located throughout the chromosome, each containing one to six silent pilin copies [8,9,10] as a source of variable genetic information. Pilin variation occurs through unidirectional DNA recombination events promoted by the RecF-like enzymes [11] that incorporate the sequence from any of the 19 silent pilin copies into *pilE*, altering the sequence of the *pilE* gene, while the silent *pilS* copy remains unchanged [9,12,13]. Pilin variation can result in a multitude of possible pilin sequences due to the many silent pilin copies and the different lengths of sequence that can be incorporated during each recombination reaction [13].

It has recently been shown that for the initiation of programmed recombination events leading to pilin antigenic variation, a G-quadruplex-forming sequence (G4 motif) located upstream of the *pilE* promoter is also required [14]. This 16-nt G4 motif d(GGGTGGGTTGGGTGGG), found in almost all of the analyzed biological probes taken from patients with gonorrhea (*N. gonorrhoeae* isolates), can fold into a parallel G4 structure in vitro [15]. It is well known that G4s formed in genomic DNA are involved in regulation of different biological processes in bacterial cells [16]. Since point mutations disrupting the quadruplex did not affect PilE synthesis [15] but blocked the pilin antigenic variation and prevented recombination, the G4 structure itself, rather than G4 motif, is important for the regulation of multiple steps of *N. gonorrhoeae* pathogenesis. Cahoon and Seifert found that the G4 motif from *N. gonorrhoeae* can form the G4 structure only during the transcription of a small non-coding RNA (sRNA) [17]. Moreover, inactivating the promoter of this *pilE* G4-associated sRNA, which is conserved in all sequenced gonococcal strains, blocks pilin variation. It is assumed that the newly synthesized sRNA forms a hybrid RNA-DNA duplex with a C-rich region, which is complementary to G4 motif, supporting the G4 formation on the opposite strand [18].

Importantly, the recombination events underlying pilin antigenic variation are also regulated by gonococcal mismatch repair (MMR) [19]. MMR is a post replicative system, conserved in prokaryotes and eukaryotes, that repairs base pair mismatches and small insertion-deletion loops that occur after DNA replication or recombination. The key proteins of MMR from *N. gonorrhoeae* are MutS and MutL, denoted as ngMutS and ngMutL, respectively [20]. ngMutS recognizes the DNA mismatch and recruits the ngMutL endonuclease, which makes a single-stranded break in the DNA daughter strand, initiating the repair process. Like other DNA repair pathways such as base or nucleotide excision repair, MMR is critically important for *N. gonorrhoeae* because it is able to correct *pilE*/*pilS* recombination events [20]. Criss et al. showed that the absence of MMR or inactivation of the ngMutS protein increased the frequency of pilin antigenic variations and, accordingly, antibiotic resistance, virulence, and adhesion to epithelial cells [19,21]. It has been suggested that *N. gonorrhoeae* mutant alleles resulting from MMR deficiency can be rapidly exchanged among bacteria, enhancing the overall survival of the population [19].

There are several hypotheses that explain the effect of MMR on the frequency of pilin variation. The most popular is the accumulation of mutations due to MMR inactivation [20]. The alternative is the involvement of MMR proteins in initiating pilin antigenic variation through interaction with the G4 structure located upstream of the *pilE* promoter [22].

Both mammalian and bacterial MutS proteins have been shown to preferentially bind to the G4 structure rather than to cognate G/T mismatch [23,24], making ngMutS a suitable candidate for regulating pilin antigenic variation. Thus, a recent study showed that ngMutS mutation leads to a 3-fold increase in pilin antigenic variability, which may be due to the loss of antirecombination properties of ngMutS or loss of G4 binding. However, according to the obtained data, the interaction of ngMutS with the *pilE* G4 structure is not essential for pilin variation [22].

In contrast, the ngMutL endonuclease binds nonspecifically to DNA, showing no preferential affinity for mismatch sites [25,26,27,28]. Nevertheless, we recently discovered the ability of this enzyme to recognize and effectively bind the single parallel G4 formed by the d(GGGT)_4_ sequence, as well as the three-quadruplex structure formed by the promoter region of the human telomerase reverse transcriptase (*hTERT*) gene; at the same time, ngMutL has been shown to not hydrolyze DNA within the tandem G4s [29].

The main goal of this study was to test the ability of the ngMutL protein to bind to the *pilE* G4 structure formed by the d(GGGTGGGTTGGGTGGGT) motif inserted into synthetic oligonucleotides. These oligonucleotides differed in the length of the G4 flanking sequences derived from the *pilE* promoter region. The obtained data will allow elucidating the role of ngMutL in the regulation of pilin variation and answering the question of whether the formation of *pilE* G4 interferes with the ATP-mediated endonuclease activity of ngMutL. The topology and thermodynamic stability of the *pilE* G4 structure in all developed DNA constructs, including control ones, was characterized using circular dichroism (CD) spectroscopy. In addition, we tried to answer the question of whether the formation of parallel G4 embedded and stabilized in a duplex context interferes with the ATP-mediated endonuclease activity of ngMutL. For this, we used G4 formed by the d(GGGT)_4_ sequence, which differs from *pilE*, d(GGGTGGGTTGGGTGGGT), by one extra T in the middle loop. The double-stranded construct containing the d(GGGT)_4_ quadruplex motif has been previously comprehensively studied in our laboratory [24].

As earlier shown, some MMR-deficient isolates of other gonococcal bacteria, *N. meningitidis* and *Pseudomonas aeruginosa*, were resistant to antibiotics. Here, we used bioinformatics tools to find a correlation between *pilE* variability, associated with drug resistance, and complete or partial deletion of DNA regions encoding ngMutS or ngMutS proteins in the genomes of *N. gonorrhoeae* isolates.

## 2. Results

### 2.1. Polymorphism Analysis of the ngmutS and ngmutL Genes in the Neisseria Genome Population

Bioinformatic analysis of the *ngmutS* and *ngmutL* genes was performed using the DNA sequences of all *N. gonorrhoeae* clinical isolates deposited into the PubMLST database. PubMLST is the largest database that stores multi-loci sequence typing data, isolate information, and an ever-increasing number of complete genomic sequences for many microbes, including *N. gonorrhoeae*. In addition, allelic variation data are available for over 2500 loci, including the core *Neisseria* genome [30,31]. This bioinformatic approach provides information on phase and antigenic variations in the *N. gonorrhoeae* population based on the number of alleles and the distribution of each mutant form in the genes of the ngMutS and ngMutL proteins in the entire population of isolates.

Inspection of *ngmutS* nucleotide polymorphisms across 15,632 *N. gonorrhoeae* isolates (Appendix A) for which data exist on the NEIS2138 locus revealed the presence of 342 allelic variants with 867 polymorphic sites for 2697 positions. Among all available isolates, large regions of the *ngmutS* gene were highly conserved (Appendix A). The most frequent allelic variants account for 13.4% of all examined *ngmutS* sequences. These data demonstrate a high level of *ngmutS* conservatism, as the variability frequency of most positions is less than 10%.

A similar analysis of *ngmutL* sequences (data exist for the NEIS1378 locus) among 15,975 *N. gonorrhoeae* isolates (Appendix A) revealed the presence of 310 variants with 460 polymorphic sites for 2017 positions. The most variable cluster located between 1370 and 1425 positions corresponds to the region of the ngMutL linker domain. However, in general, the *ngmutL* genes still have a low frequency of variations in most positions, equal to 14.2% for the entire population of isolates (Appendix A).

Given that the PubMLST database contains information on *N. gonorrhoeae* isolates with unassigned alleles, we verified genomes of these isolates to analyze the presence or absence of the *ngmutS* and *ngmutL* genes independently of each other (Appendix A). A total of 432 isolates were found to contain the *ngmutS* gene but for which the *ngmutS* allele was not assigned, and 15 isolates were *ngmutS*-deficient (Appendix A). In the same way, 1016 isolates were identified with the unassigned *ngmutL* allele but carrying the gene of interest. Only nine *N. gonorrhoeae* isolates lacked the *ngmutL* gene (Appendix A).

### 2.2. Correlation between N. gonorrhoeae MMR Deficiency and Frequency of Pilin Antigenic Variation

To study the effect of *ngmutS*- or *ngmutL*-deficient *N. gonorrhoeae* strains on the frequency of pilin variation, we divided all isolates to several groups: (1) containing the *ngmutS* or *ngmutL* gene with the assigned allele; (2) containing the *ngmutS* or *ngmutL* gene with unassigned allele; (3) without the gene of interest. Depending on the type of gene—*ngmutS* (S) or *ngmutL* (L), these groups were designated as 1S or 1L, 2S or 2L, and 3S or 3L, respectively. We analyzed the *pilE* mutation frequencies in each group, but because *pilE* genes were low-conserved, *pilE* alleles were not assigned in most genomes isolate populations from *N. gonorrhoeae*.

Due to this limitation of the PubMLST database, we used BLAST (The Basic Local Alignment Search Tool) and the *pilE* sequence of *N. gonorrhoeae* strain MS11 as a consensus to retrieve the *pilE* gene from all isolates. After the *pilE* sequences were aligned in each isolate group using the Jalview program, we calculated the pilin variation frequency. The total number of *pilE* polymorphic sites in each group is presented in Figure 1A.

It was shown that the absence of the *ngmutS* or *ngmutL* genes in the 3S and 3L groups of *N. gonorrhoeae* isolates increased the mutation frequency in *pilE* (>25%) compared to other studied groups (Figure 1B). In addition, unlike other groups, the proportion of low-variability positions decreased in 3L, but not in 3S. Interestingly, only two *N. gonorrhoeae* isolates lack both *ngmutS* and *ngmutL* sequences.

We determined the average variability of *pilE* polymorphic sites in each group of *N. gonorrhoeae* isolates as the proportion of isolates whose *pilE* gene contains a substitution at the polymorphic position. The mean value of variable positions in groups 1S and 1L was 0.125 and 0.14, respectively. In groups 2S and 2L, these values remained almost unchanged (0.142 and 0.151, respectively), while in groups 3S and 3L, they increased to 0.251 and 0.255, respectively. According to statistical analysis, Student’s *t*-test revealed a statistically significant difference in mean variability between groups 1S and 3S (*p* << 0.05), 1L and 3L (*p* << 0.05), 2S and 3S (*p* << 0.05), 2L and 3L (*p* << 0.05), but not between 1S and 2S (*p* = 0.08) and 1L and 2L (*p* = 0.25).

Genomic mutations do not always correlate with amino acid substitutions. In order to assess amino acid polymorphisms and select conserved and variable positions of the *N. gonorrhoeae* PilE protein, we performed antigenic polymorphism mapping on the crystal structure of *N. gonorrhoeae* PilE (PDB:2HI2) solved by Parge et al. [32] (Figure 2). The degree of conservatism at a particular amino acid position was quantified using numerical indexes in the Jalview software (see Materials and Methods, Section 4.1.4, Appendix A). Using the PyMol protein structure visualization program, we indicated the conserved and variable regions of PilE amino acid sequences in groups of *ngmutS-* or *ngmutL*-deficient *N. gonorrhoeae* isolates (Figure 2A and Figure 2B, respectively).

Almost all hypervariable PilE positions from both *ngmutS*- and *ngmutL*-deficient groups are located in the surface-exposed areas within the D-region between two conserved cysteine residues (marked with sticks and highlighted in red in Figure 2) [33] and in the αβ loop. Low prevalence polymorphisms are scattered throughout the β-sheet domain and are shown in blue and gray for *ngmutS*- and *ngmutL*-deficient groups of *N. gonorrhoeae* isolates, respectively.

### 2.3. The Secondary Structure of the Engineered Single- and Double-Stranded DNA Models

To investigate the role of *pilE* G4 in *N. gonorrhoeae* MMR-mediated regulation of pilin variation, we examined the binding affinity and functional response of ngMutL to the quadruplex structure. Although ngMutS is a known G4 binder, it has been shown that the effect of ngMutS inactivation is not related to the loss of its binding to *pilE* G4 [22]. In addition, the ngMutL protein is much less studied than ngMutS and has an endonuclease function that allows monitoring the efficiency of DNA cleavage. It is known that MutL has a weak DNA-binding activity, which largely depends on the DNA length [34]. The size of the DNA region bound by MutLs of different species varies from 30 to 250–500 bp [35]. Interactions of ngMutL with single- and double-stranded DNA fragments of various lengths were studied [26]. This protein was shown to be able to efficiently form complexes with DNA fragments that are longer than 40 nucleotides [26]. Only for MutL from *E. coli* the effective binding with G4 inside of duplex structure was demonstrated in our earlier study [24]. The ability of ngMutL to effectively bind a single G4 structure remains unknown.

We used synthetic DNA oligonucleotides of various lengths containing the *pilE* G4 motif flanked by the same sequences as in the *N. gonorrhoeae* genome, which are presented in Table 1 along with complementary oligomers (marked by symbol M) and respective control models. The DNA duplexes formed by hybridization of the oligonucleotides listed in Table 1 and their abbreviations are shown in Figure 3.

Using CD spectroscopy, we examined the folding topology and thermal stability of the G4 structures formed in the 19pilG4, 41pilG4, and 95pilG4 oligonucleotides, which differ in the length of the G4 flanks. The CD spectra of these single-stranded DNAs in buffer solution containing 5 mM KCl revealed a typical G4 parallel fold with a positive peak at 265 nm and a negative one at about 240 nm (Figure 4). From CD data recorded at different temperatures, melting curves were obtained; a reduced KCl concentration was chosen to capture the entire G4-coil conformation transition within the available temperature range.

The observed differences between the sets of CD spectra of oligonucleotides having various lengths are explained by the increasing influence of single-stranded unstructured sequences flanking the quadruplex structure during the transition from 19pilG4 to the 95pilG4. The same factor determines the drop in the melting temperature (T_m_) from more than 77 ± 1 for 19pilG4 to 66 ± 1 for 41pilG4 and then to 63 ± 1 °C for 95pilG4 due to an increase in the unfavorable entropy contribution. As expected, the addition of the complementary strand (95M) to 95pilG4 causes dramatic changes in the CD spectrum, which displays a peak near 280 nm, where the signals of unstructured oligonucleotides and B-DNA with an average G/C content are located (Figure 4D). This long oligonucleotide forms a duplex structure at 37 °C even in the buffer containing 100 mM KCl, which promotes G4 folding. Regarding the structure formed by the shorter 41pilG4 and fully complementary DNA strand (41M), the positive CD band at 270 nm can be assigned to either a G-rich DNA duplex or a duplex with a partially preserved quadruplex structure (Figure 4D). A number of studies have shown that the G4–DNA duplex equilibrium depends on many factors that affect the relative stability of both secondary structures, in particular, on the length of G4 flanking sequences; it has been demonstrated that the propensity to attain a G4 structure decreases with increasing length of the flanks [36]. Accordingly, to create a 19 bp duplex, we had to introduce four G→T substitutions into the extremely stable 19pilG4 to prevent G4 formation. The resulting 19T was hybridized with 19M_A to give a 19T/19M_A duplex structure (Figure 4D).

The folding capacity and thermal stability of both G4 (formed by the d(GGGT)_4_ sequence) and duplex domains in the 95G4/76M construct containing a parallel G4 stabilized in a DNA duplex context (Figure 3) have been previously described and analyzed by a variety of biophysical and biochemical techniques [24].

### 2.4. Comparative Binding of ngMutL to Single-Stranded DNA Fragments of Different Lengths Containing pilE G4 and Its Double-Stranded Analogues

Since the DNA-binding activity of MutL, one of the principal components of the mismatch repair pathway, is highly dependent on DNA length, we evaluated the effect of *pilE* G4 on recognition and binding to ngMutL using the synthetic oligonucleotides 19pilG4, 41pilG4, and 95pilG4, which differ in the length of quadruplex-flanking regions (Table 1). Double-stranded versions of these DNA fragments, stabilized by 19, 41, and 95 base pairs, respectively (Figure 3), were used as related ngMutL substrates. Complex formation between the ngMutL protein and the entire set of DNA ligands bearing the TAMRA fluorophore at the 3′ end of the oligonucleotide strand (indicated in Table 1) was monitored by electrophoretic mobility shift assay (EMSA). Figure 5 shows binding curves of ngMutL to 41- and 95-nt DNAs containing *pilE* G4 and the corresponding duplex structures of the same length in dependence on the total protein concentration, as well as the values of apparent dissociation constants (*K*_d_^app^) calculated from EMSA data, and a representative polyacrylamide gel images of ngMutL binding to 95pilG4 and 41pilG4.

As expected, the complex formation of ngMutL with the longest oligonucleotide 95pilG4 was significantly more efficient than with 41pilG4; their *K*_d_^app^ values differ by almost a factor of three (Figure 5B), although the quadruplex structure in both DNA ligands is the same. In the case of 41pilG4 binding to ngMutL, two bands were observed at high protein concentrations (Figure 5D), which correspond to complexes formed by ngMutL oligomers. Importantly, this pattern is maintained for the respective double helices (compare the *K*_d_^app^ values, 123 and 650 nM, for the 95- and 41-bp duplexes in Figure 5B). In addition, comparison of *K*_d_^app^ values obtained for quadruplex-containing and duplex ligands revealed that ngMutL binds to parallel *pilE* G4 with higher affinity than double-stranded DNA, regardless of ligand length; that is, this conclusion is true for both 95- and 41-mer DNAs.

It should be noted that we were unable to calculate the *K*_d_^app^ value for ngMutL binding to the 19pilG4 oligonucleotide due to a dramatic decrease in ngMutL affinity to this short DNA ligand [26]. However, the formation of a nucleic acid–protein complex (~30%) was observed at a 25-fold excess of the ngMutL dimer relative to 19pilG4 (Figure 6A). At the same time, no interaction of ngMutL with double-stranded 19T/19M_A was observed under the same binding conditions (Figure 6A). In order to evaluate the efficiency of the ngMutL complex formation with 95-, 41-, and 19-mer DNA ligands (both single- and double-stranded) and to compare the effect of ligand length and composition (absence or presence of the G4 structure) on this process, we used the same conditions, which are presented in the legend of Figure 6.

It has been shown that while the yields of nucleic acid–protein complexes 19pilG4 and 19T/19M_A with ngMutL (formed with the participation of 19-mer single-stranded and double-stranded DNA ligands) range from 30 to 0%, respectively, they vary from almost 80 to 40% for ngMutL complexes with 41pilG4 and 41pilG4/41M (41-mer single- and double-stranded DNA ligands). Interestingly, there was no significant difference in the formation of the ngMutL complex with 95pilG4 and its double-stranded analogues; their yields reached 100% (Figure 6B). These data clearly confirm the preferential binding of ngMutL to the G4 structure compared to the DNA duplex, on the one hand, and allow us to estimate the size of the double helix region occupied by the ngMutL protein, on the other hand. According to our data, the size of the ngMutL loading site exceeded 41 base pairs but was less than 95 pairs. This is consistent with the previously published finding that ngMutL is able to efficiently form complexes with DNA fragment greater than 40 base pairs in length [26].

### 2.5. ngMutL-Mediated Cleavage of the G4-Containing Double-Stranded DNA Substrate

In eukaryotes and most bacteria, MutL, which uses a methyl-independent mechanism, works as the endonuclease, nicking the newly synthesized DNA strand, thus activating the mismatch correction. As other MutL homologous, ngMutL is a homodimer and consists of C- and N-terminal main domains, which are responsible for DNA cleavage and binding, correspondingly [37]. It was previously shown that the binding of the β-subunit of DNA polymerase III (β-clamp) by the endonuclease domain of ngMutL promotes nicking activity towards the nascent DNA strand [37].

Recently, we have tested the ngMutL ability to cleave the single-stranded DNA containing a tandem three-quadruplex structure of the *hTERT* promoter and revealed that it is significantly suppressed by the stable G4 scaffold [29]. Herein, we evaluated the efficiency of ngMutL-mediated cleavage of the single parallel G4 formed by the d(GGGT)4 sequence, which is stabilized in a DNA duplex context due to the lack of the site complementary to the G4-forming motif in the opposite strand, (95G4/76M). Of note, the G4 motif under investigation is very close to that forming *pilE* G4, differing in only one extra T residue in the middle loop. For comparison, a 76 bp B-DNA fragment with a random sequence (76R/76M) (Figure 3) was used as a cognate ngMutL substrate. The endonuclease activity of ngMutL was tested in the presence of equimolar amounts of β-clamp from *N. gonorrhoeae* (ngβ), as described earlier [29,38]. The ngMutL-mediated DNA nicking resulted in shorter 3′-TAMRA-labeled oligonucleotide cleavage products, which were separated from the intact DNA by gel electrophoresis in denaturing conditions, enabling an evaluation of the cleavage efficiency and the product length. The hydrolysis of 76R/76M was revealed to be rather efficient and non-specific (Figure 7), while the cleavage extent of the 95G4/76M was markedly reduced compared to the 76-bp duplex lacking the G4 structure (Figure 7). In addition, the nicks introduced by the ngMutL protein were located before and after the G4 motif, but not inside the quadruplex structure.

As can be seen from Figure 7, the conditions we used did not allow us to achieve complete hydrolysis of even the canonical duplex 76R/76M. However, it was important to study the peculiarities of ngMutL-induced hydrolysis of the G4-containing substrate, when the extent of the cleavage, at least 76R/76M, reaches 100%.

At the first stage, the optimal DNA:ngMutL ratio of 1:50 was found, which provides 100% endonuclease activity on the 76R/76M substrate. This process is accompanied by the accumulation of short DNA cleavage products (no longer than 14 nucleotide units) (Figure 8).

It is well known that the MutL endonucleases have different nicking efficiency in the presence of various metal ions [25,27,39,40,41,42]. It was previously shown on plasmid substrates that Mn^2+^, Mg^2+^, and Ca^2+^ activate the endonuclease activity of ngMutL [25]. Here, we tried to use different metal ions to optimize the hydrolysis of linear DNA 76R/76M with ngMutL in the presence of ngβ. We have showed that ngMutL exhibits the highest endonuclease activity in the presence of Mn^2+^, similar to other bacterial MutL homologs [27]. Mg^2+^ also activates the ngMutL endonuclease function (Appendix A), but less efficiently than Mn^2+^. Interestingly, unlike the previous study, Ca^2+^ does not support ngMutL-mediated cleavage of the 76-bp duplex; this is also true for Cd^2+^, Co^2+^, Ni^2+^, and Zn^2+^. The addition of Mg^2+^ or Ca^2+^ to the reaction mixture does not significantly affect the ngMutL activity. Thus, the Me^2+^ conditions used in previous studies—a combination of 5 mM MgCl_2_ and 5 mM MnCl_2_—are optimal [29,38]. Mg^2+^ is required for ATPase activity of MutL, and Mn^2+^ is needed for endonuclease function [43].

We then examined the effect of ATP on the ability of ngMutL to nick double-stranded DNA substrates, including the G4-containing one, at a DNA:ngMutL ratio of 1:50. ATPase activity of MutL is known to be required for DNA repair processing. The addition of 0.5 mM ATP already increased nicking efficiency mediated by an equimolar mixture of ngMutL and ngβ in both the 76R/76M and 95G4/76M DNA duplexes compared to the conditions described in the legend of Figure 7 (0.8 mM ATP and DNA:ngMutL ratio of 1:25). Moreover, the cleavage products are different for DNA substrates in the presence and in the absence of ATP (Figure 9). Additionally, 0.5 mM ATP in the reaction mixture causes almost complete cleavage of 76R/76M, and the length of the hydrolysis products as a whole does not exceed 20 nucleotides (Figure 9).

Probably, during ATP hydrolysis, conformational changes in ngMutL allow the enzyme to dissociate from DNA. Accordingly, truncated (already cleaved) DNA fragments can be rehydrolyzed by ngMutL until the DNA length matches the length of the ngMutL loading site, which is more than 41 base pairs (nucleotides). However, the cleavage products are much shorter, which indicates not a simple dissociation of the enzyme from DNA during ATP hydrolysis, but rather “sliding” along the substrate.

In the absence of ATP, we observe predominantly long cleavage products (at least 60 nucleotides), and the efficiency of 76R/76M cleavage is much less than 100%. Under these conditions, ngMutL forms a strong complex with DNA that prevents the dissociation of the enzyme and(or) sliding along the substrate [34] and, therefore, the appearance of short cleavage products (Figure 9).

It should be noted that a similar effect was also observed for the G4-containing DNA substrate 95G4/76M. Without ATP, long cleavage products are accumulated. During ATP hydrolysis, the conformational changes in ngMutL-ngβ lead to the appearance of short cleavage products and an increase in nicking efficiency, but ngMutL does not nick DNA in the region of G4 formation. This is consistent with the previously published data that ngMutL is unable to hydrolyze DNA inside the three-quadruplex structure formed by the promoter region of the *hTERT* gene [29].

Excess ATP (5 mM) significantly inhibits 76R/76M cleavage, possibly through chelation of Mg^2+^ and Mn^2+^.

To understand the reasons for the ATP-induced stimulation/decrease of ngMutL endonuclease activity, we tested the effect of ATP on the DNA–protein complex formation using the same DNA substrates (76R/76M and 95G4/76M). Appendix A shows that increasing ATP concentration significantly reduces the ability of ngMutL to bind the DNA duplex 76R/76M, but not G4-containing 95G4/76M. This once again indicates a higher affinity of ngMutL to G4, which is less regulated by ATPase activity than the affinity to the canonical DNA duplex.

## 3. Discussion

Bacterial pili are long surface structures classified based on phenotypic and molecular characteristics [44]. Type IV pili are multifunctional nanofibers present on the surface of numerous bacteria that play a role in surface motility, biofilm formation, adhesion to human epithelial cells, and virulence of pathogens such as *N. gonorrhoeae*. Type IV pili include several proteins, but the fibers are primarily composed of a single subunit called the major pilin or PilE. The phase and antigenic variation of the *N. gonorrhoeae* pilin is an important virulence factor associated with antibiotic resistance. The hyper variability of the pilin region accessible to human antibodies ensures the survival and pathogenesis of bacteria.

The process of pilin (PilE) antigenic variation, realized through homologous recombination [45], is partially controlled by mismatch repair pathway [19,22]. Using a bioinformatics approach, we have for the first time found a correlation between *pilE* variability and complete or partial deletion of DNA regions encoding ngMutS or ngMutL proteins in the genomes of all *N. gonorrhoeae* isolates deposited in the PubMLST database. Despite the large number of polymorphic sites, both genes are conserved; *ngmutS* has no significant substitutions, while *ngmutL* contains a variable linker region (Appendix A). Among more than 15,000 isolates, we found only 15 and nine, whose genomes lacked the *ngmutS* and *ngmutL* genes, respectively. It was shown that the absence of the *ngmutS* or *ngmutL* genes in the 3S and 3L groups of *N. gonorrhoeae* isolates increased the mutation frequency in *pilE* (>25%) compared to other studied groups (Figure 1B). According to the structural mapping of *N. gonorrhoeae* PilE (Figure 2), almost all hypervariable PilE positions from both *ngmutS*- and *ngmutL*-deficient groups of *N. gonorrhoeae* isolates are located in the surface-exposed areas.

It has recently been shown that the G4 motif located upstream of the *pilE* promoter is required to initiate recombination events leading to pilin antigenic variation [14]. Endogenous G4s are found in certain G-rich sequences and result from self-association of guanine residues to form stacked G-tetrads that are stabilized by Hoogsteen hydrogen bonds and interactions with metal ions (mainly K^+^) that are coordinated in the central cavity. G4 structures are highly polymorphic; they can adopt parallel, antiparallel, and hybrid (3 + 1) topologies characterized by different orientation of four G-tracts in the quadruplex core [46]. As structural elements of the genome, G4s are recognized by numerous cellular proteins and enzymes and are involved in the regulation of various chromosome functions, such as DNA replication, transcription initiation [47], chromosome end protection, DNA repair [48] and recombination [15,17], and epigenetic genome regulation [49,50]. One of several hypotheses explaining the effect of MMR on the frequency of pilin variation is the regulation of homologous recombination through the interaction of MMR proteins with the G4 structure associated with the *pilE* promoter [22]. While G4-binding activity is well known for ngMutS [20], it has not previously been shown for ngMutL. The main goal of our study was to examine the ability of the ngMutL protein to specifically bind *pilE* G4 and to understand whether the G4 structure affects the recombination-associated endonuclease activity of ngMutL.

To model the *pilE* G4 structure, we constructed several synthetic d(GGGTGGGTTGGGTGGGT)-containing single-stranded DNAs that differed in length (Table 1). According to CD spectroscopy data, 19-, 41-, and 95-nt DNA fragments are folded into parallel G4 structures, whose thermodynamic stability depends on the length of the sequences flanking the quadruplex structure; the T_m_ values decreased with their lengthening due to unfavorable entropy contribution (Figure 4A–C). The same factor determined the formation of the DNA duplex structure by hybridization of 41pilG4 and 95pilG4 with fully complementary oligonucleotides (Figure 4D). Since the equilibrium G4–DNA duplex depends on the length of G4 flanking sequences [36], we had to introduce four G→T substitutions in the extremely stable 19pilG4 to prevent G4 formation and stabilize the 19 bp duplex (Figure 3 and Figure 4D).

Binding of ngMutL to single-stranded DNA fragments of different lengths containing *pilE* G4 and its double-stranded analogs was monitored by EMSA assay (Figure 5 and Figure 6). As expected, the most effective formation of the nucleic acid –protein complex is observed for the longest oligonucleotide 95pilG4 and the related duplex; the binding activity decreased with the shortening of the *pilE* G4-containing and duplex ligands. However, the obtained data convincingly confirmed the preferential binding of ngMutL to the G4 structure compared to the DNA duplex, although the latter is a cognate substrate for the ngMutL endonuclease. This finding correlates with our previous data obtained for MutL protein from *E. coli* [24].

The endonuclease activity of ngMutL was tested in the presence of equimolar amounts of β-clamp from *N. gonorrhoeae*. As double-stranded DNA substrates, we used parallel G4 formed by the d(GGGT)_4_ sequence, which is stabilized in a DNA duplex context due to the lack of the site complementary to the G4-forming motif in the opposite strand (95G4/76M), and ordinary 76 bp DNA duplex 76R/76M (Figure 3). We recently tested the ngMutL ability to cleave single-stranded DNA containing the tandem three-quadruplex structure of the *hTERT* promoter and found that it was significantly suppressed by the stable G4 scaffold [29]. As evident from Figure 8, the cleavage extent of 95G4/76M was markedly reduced compared to the 76 bp duplex lacking the G4 structure, with the nicks introduced by the ngMutL located before and after the G4 motif, but not inside the quadruplex structure. Thus, ngMutL does not nick DNA in the region of G4 formation but is able to bypass this noncanonical structure.

Among the factors influencing the efficiency and direction of ngMutL-induced DNA cleavage, we studied the effect of various metal ions (Appendix A), DNA:protein ratio and ATP cofactor (Figure 9 and Appendix A). Optimization of the reaction conditions, which increased the efficiency of ngMutL-mediated cleavage of DNA substrates up to 100%, did not change our main conclusion: the G4 structure formed via the conformational rearrangement of the G-rich region of genome DNA inhibits ngMutL-induced DNA nicking and modulates cleavage positions. A possible role of the G4 in the regulation of DNA nicking may involve the sequestration of ngMutL at the G4 structure. Our findings led us to speculate that *pilE* G4 may regulate the efficiency of pilin antigenic variation by G4 binding to ngMutL and suppressing the homologous recombination process.

## 4. Materials and Methods

### 4.1. Bioinformatic Analysis of the Correlation between N. gonorrhoeae MMR Deficiency and the Frequency of Pilin Antigenic Variation

#### 4.1.1. Allele Mining of *ngmutS* and *ngmutL*

For the loci of the *ngmutS* and *ngmutL* genes, as well as further analysis of the alleles, we used the PubMLST plugin according to the resource [31]. Using the *ngmutS* and *ngmutL* gene sequences of the *N. gonorrhoeae* strain MS11 (NCBI: WP_003688085.1 and WP_003688688.1) as a consensus, the NEIS2138 and NEIS1378 loci, respectively, were found to represent the nucleotide sequences. All assigned *ngmutS* and *ngmutL* alleles of *N. gonorrhoeae* isolates presented in the database were collected and listed in Appendix A, respectively. The PubMLST plugin was used to analyze the polymorphisms of the collected *ngmutS* and *ngmutL* alleles. Next, the nucleotide sequences of these alleles were exported to FASTA format for translation into protein sequences and alignment in Jalview using the ClustalW algorithm.

#### 4.1.2. Gene Presence Analysis

To search for *N. gonorrhoeae* isolates that lack the *ngmutS* or *ngmutL* gene in their genome, the PubMLST “*Gene presence*” plugin was used. In the “*Loci*” list, the NEIS2138 or NEIS1378 gene loci for *ngmutS* or *ngmutL*, respectively, were selected. “*Parameters and options*” were not changed: the minimum percentage of identity and alignment was 70% and 50% of the total alignment length, respectively. The result is displayed in the form (Appendix A), in which the first column is the id of the isolate, the second is “1” or “0” if the gene is present or absent, respectively. All ids of isolates in the genome of which the *ngmutS* or *ngmutL* gene was present were collected and further analyzed using the *“Dataset”* plugin to find those isolates in which the allele was not assigned.

#### 4.1.3. The Effect of the *N. gonorrhoeae* MMR Deficiency on the Frequency of Pilin Antigenic Variation

The analyzed isolates of *N. gonorrhoeae* were divided to three groups (Appendix A): (1) containing the *ngmutS* or *ngmutL* gene with the assigned allele; (2) containing the *ngmutS* or *ngmutL* gene with unassigned allele; (3) without the gene of interest. Depending on the type of gene—*ngmutS* (S) or *ngmutL* (L), these groups were designated as 1S or 1L, 2S or 2L, and 3S or 3L, respectively. We analyzed the variability of the *pilE* in each group using the “*BLAST*” plugin of the PubMLST database. The *pilE* nucleotide sequence of MS11 *N. gonorrhoeae* was used as a consensus (UniProtKB/Swiss-Prot: P02974.2). Next, the nucleotide sequences of the *pilE* gene of each group were exported to FASTA for alignment using the ClustalW algorithm. The alignment results were re-stored in Jalview and analyzed in Python to define the frequency of variations for each *pilE* position. The output TXT files after calculation were converted into Excel tables containing information about *pilE* polymorphisms for further analysis. A site of polymorphism is the position, the nucleotide of which differs in the gene of at least one isolate among all analyzed isolates. Conservative positions are positions whose nucleotide is retained more often than in 90% of cases among the genomes of all analyzed isolates. Variable positions are those positions whose nucleotide is retained less frequently than in 75% of cases among the genomes of all analyzed isolates. Statistical analysis (Student’s *t*-test) was performed to identify a statistically significant difference in mean variability between groups.

#### 4.1.4. Antigen Polymorphism Structural Mapping

We translated nucleotide *pilE* alignment of 3S and 3L groups in Jalview to map the conservative or variable positions to the crystal structure of the protein PilE from the *N. gonorrhoeae* MS11 strain (PDB: 2HI2). Jalview quantifies conservative positions using numerical indices from 0 to 9: the higher the value, the more identical the substitutions in the sequence [51]. Numerical indices show an automatically calculated quantitative alignment annotation, which measures the conservatism of residues. This calculation is based on the AMAS method of multiple sequence alignment analysis. The designations “*” and “+” show the complete conservatism of the residue in the sample and the presence of a small number of substitutions that have absolutely no effect on the physicochemical properties, respectively. We used PyMol2 (version 2.4.1, 2020-09-14, Schrödinger Inc., New York, NY, USA) to visualize the structure. To mark variable residues, all amino acids were selected, the numerical indices of which were less than 4, and they were stained yellow. For marking conservative residues, we selected all of the amino acids with “*” or “+” designations and colored them blue or gray for PilE from 3S or 3L groups, respectively. The remaining protein residues with numerical indices from 4 to 9 are colored light gray.

### 4.2. Oligodeoxyribonucleotides

All oligodeoxyribonucleotides (synthesized via standard phosphoramidite chemistry and purified by high-pressure liquid chromatography in Syntol, Russia) were used without further purification. Oligonucleotide strand concentrations were determined spectrophotometrically using extinction coefficients derived from the nearest-neighbor data (https://www.idtdna.com/calc/analyzer, accessed on 10 December 2022).

### 4.3. Preparation of Intramolecular G4 Structures and DNA Duplexes

G4s were prepared by annealing the 95-, 41-, or 19-nt single-stranded DNA samples, including those labeled with tetramethylrhodamine (TAMRA), by heating them at 95 °C for 3 min and cooling overnight to 4 °C in appropriate buffer solutions.

To obtain DNA duplexes, complementary DNA strands were annealed (by heating at 95 °C for 5 min and slowly cooling to room temperature) in an appropriate buffer solution; the unlabeled strand was used in a 5% excess compared to the TAMRA-labeled strand. The composition of buffers is presented in Figure legends.

### 4.4. Purification of Recombinant Proteins

Protein ngMutL was purified as described previously [25]. The recombinant ngMutL expressed as N-terminal His_6_-tagged proteins in the *E. coli* strain BL21(DE3) and purified by Ni-NTA affinity chromatography. Purification was followed by size exclusion chromatography on a Superdex 200 TM 10/300 (GE Healthcare, Chicago, IL, USA) on an Äkta Purifier (GE Healthcare, Chicago, IL, USA). The resulting proteins were aliquoted, frozen in liquid nitrogen, and stored in 10 mM HEPES-KOH buffer (pH 7.9) with 300 mM KCl, 1 mM EDTA, 10% (*v*/*v*) glycerol, and 1 mM 2-mercaptoethanol at −80 °C.

The recombinant β-subunit of DNA polymerase III (ngβ) was purified similarly to ngMutL by Ni-NTA affinity chromatography. The ngβ sample was further purified on HiTrap Heparin HP (1 mL) column pre-equilibrated in 10 mM K-phosphate buffer (pH 7.5) with 0.5 mM EDTA. The protein was eluted with a salt gradient to 0–0.5 M KCl as previously described [52]. The ngβ fractions were dialyzed in a 10 mM HEPES-KOH buffer (pH 7.9) with 200 mM KCl, 1 mM EDTA, and 10% (*v*/*v*) glycerol; then, they were aliquoted, frozen in liquid nitrogen, and stored at −80 °C.

Protein concentration in the fractions was determined spectrophotometrically with a NanoDrop ND-1000 (Thermo Fisher Scientific, Waltham, MA, USA) at 280 nm. Extinction coefficients were calculated with ProtParam Tool (https://web.expasy.org/protparam, accessed on 15 January 2023).

### 4.5. Circular Dichroism Measurements

Oligonucleotides containing the G4 motif (Table 1) were annealed in 20 mM HEPES buffer (pH 7.3) containing 5 mM KCl and 140 mM NaCl to enable G4 conformation. Double-stranded DNA probes (Figure 3) were annealed in 20 mM HEPES buffer (pH 8.0) containing 100 mM KCl according to the standard procedure. CD spectra were recorded in a quartz cuvette of 10 mm optical path length between 30 and 85 °C in temperature intervals of ~5 °C at the average heating rate of 0.5 °C/min on a Chirascan CD spectrometer (Applied Photophysics Ltd., Leatherhead, UK) equipped with a Peltier controller. The DNA concentration was chosen to attain absorption of 0.4–0.6 at 260 nm, which yields an optimum signal-to-noise ratio. The measurements were performed in the 230–360 nm wavelength range at a scanning speed of 30 nm/min and a signal averaging time of 2 s with constant flow of dry nitrogen. All the CD spectra were baseline-corrected for signal contributions caused by the buffer. CD spectra were plotted as a molar CD per oligonucleotide strand against wavelength. The spectra were processed with the Origin 8.0 software (Electronic Arts, Redwood City, CA, USA) using the Savitzky–Golay filter. The CD melting profiles revealed the temperature dependence of the CD signal at 265 nm.

### 4.6. DNA-Binding Activity of ngMutL

TAMRA-labeled DNA probes (20 nM) were incubated for 10 min on ice with 15–1000 nM ngMutL (per dimer) in 20 mM HEPES buffer (pH 8.0) containing 100 mM KCl, 0.5 mg/mL BSA, and 1 mM DTT. Samples (10–20 μL) were analyzed by electrophoresis in a non-denaturing 6% polyacrylamide gel in TAE buffer for 2–3 h at 4 °C. Gel photographs obtained on a Typhoon FLA 9500 device (GE Healthcare, Chicago, IL, USA) were processed using the TotalLab TL120 software (Nonlinear Dynamics Ltd., New Castle, UK). The intensity of the zones corresponding to free DNA and the complex of MutL protein with DNA was measured. The extent of binding was calculated as the ratio of the intensity of the zone corresponding to the DNA –protein complex to the total intensity of the zones, multiplied by 100%. The *K*_d_^app^ values of ngMutL complexes with DNA was determined as the protein concentration at which 50% of the DNA ligand was in a complex with the enzyme. The calculation was carried out for at least three independent experiments.

### 4.7. Hydrolysis of DNA by ngMutL

The ngMutL endonuclease activity was measured as described previously [38]. Reaction mixtures (10 μL) contained 20 mM HEPES-KOH buffer (pH 8.0), 100 mM KCl, 5 mM MgCl_2_, 5 mM MnCl_2_, 0.8 mM ATP, 0.5 mg/mL BSA, 12% (*v*/*v*) glycerol, 10 nM 3′-TAMRA-labeled double-stranded DNA (Table 1), and 250 nM ngMutL (per dimer) in the presence of equimolar amounts of the ngβ dimer. Samples were incubated at 37 °C for 90 min and quenched by the addition of 50 mM EDTA and 1 mg/mL proteinase K followed by incubation at 55 °C for 20 min. The products were analyzed by electrophoreses in a 12% polyacrylamide gel containing 7 M urea. Gels were visualized using a Typhoon FLA 9500 (GE Healthcare, Chicago, IL, USA). The analysis of hydrolysis products was carried out using the TotalLab TL120 software (Nonlinear Dynamics Ltd., New Castle, UK).

Various conditions were used to achieve 100% cleavage of double-stranded 76R/76M (Figure 3). Reaction mixtures (10 μL) contained 20 mM HEPES-KOH buffer (pH 8.0), 100 mM KCl, 0.5 mg/mL BSA, 12% (*v*/*v*) glycerol, 10 nM 3′-TAMRA-labeled double-stranded DNA 76R/76M (Table 1); 5 mM CaCl_2_ or other metal-ions (5 mM CdCl_2_, 5 mM CoCl_2_, 5 mM NiCl_2_, 5 mM ZnCl_2_, 5 mM MgCl_2_, and/or 5 mM MnCl_2_), 0.2–0.8 mM or 5mM ATP, and 500–750 nM ngMutL (per dimer) in the presence of equimolar amounts of the ngβ dimer.

## 5. Conclusions

The host-adapted human pathogen *N. gonorrhoeae*, like many other bacteria, uses homologous recombination to undergo phase and antigenic variations and evade the immune system. The surface protein pilin (PilE), the predominant component of type IV pili that controls attachment to the human cell, is one of the targets for antigenic variation [15]. It has recently been shown that the recombination events underlying pilin antigenic variation may be regulated by the *N. gonorrhoeae* MMR pathway [19], as well as by the G4 structure located upstream of the *pilE* promoter. The main participants in the methyl-independent *N. gonorrhoeae* MMR are the ngMutS and ngMutL proteins. Using bioinformatics tools, we analyzed the genomes of *N. gonorrhoeae* isolates deposited in the PubMLST database. It has been established for the first time that the loss of *ngmutS* or *ngmutL* genes leads to an increase in *pilE* variability in the cells of pathogen isolates. We then focused on the understudied ngMutL having an endonuclease function, which could be associated with recombination events. An increased affinity of ngMutL to *pilE* G4 compared to the DNA duplex was shown. Nevertheless, the enzyme does not cleave the DNA inside G4, but is able to bypass this noncanonical structure. A possible mechanism of the G4-mediated inhibition of DNA nicking may include the sequestration of ngMutL at the G4 structure.

These findings have implications both for understanding the mechanism of *N. gonorrhoeae* antigenic variation and its role in microbial pathogenesis.

## Figures and Tables

**Figure 1 ijms-24-06167-f001:**
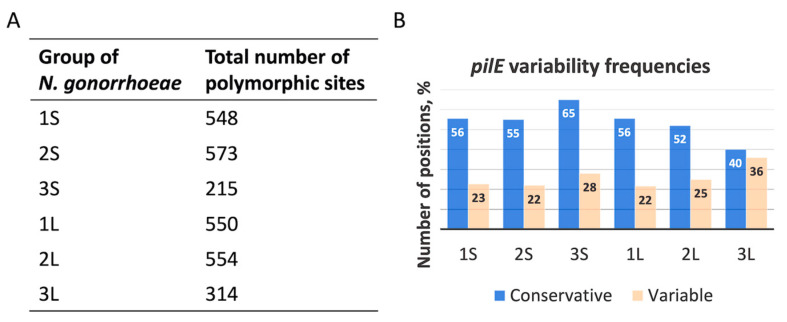
Polymorphism of *pilE* in groups of all *N. gonorrhoeae* isolates. Total number of *pilE* polymorphic sites (**A**). Number of *pilE* positions with specific nucleotide substitutions in each group of *N. gonorrhoeae* isolates as the ratio of conservative (<10%) or variable (>25%) substitutions to the total alignment length (**B**). Data are presented as a percentage.

**Figure 2 ijms-24-06167-f002:**
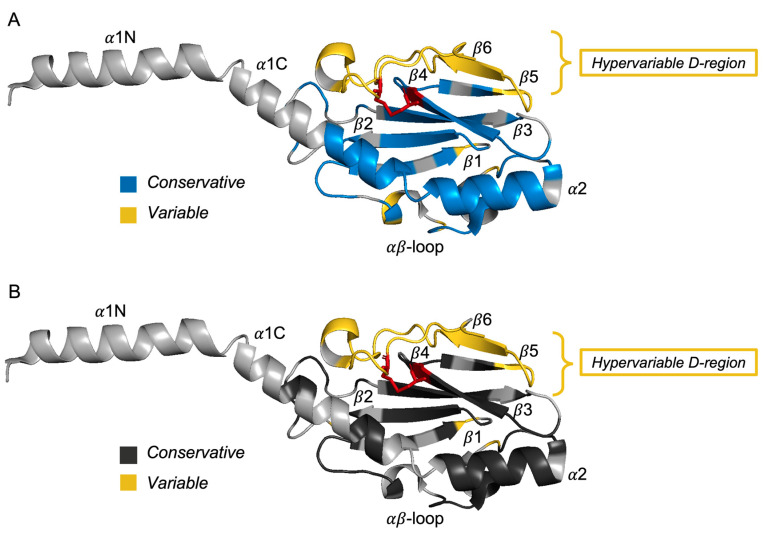
Structural mapping of *N. gonorrhoeae* PilE (PDB: 2HI2) conserved and variable amino acid positions corresponding to Jalview numerical indexes. Conservative positions are shown in blue or dark gray for *ngmutS*- (**A**) and *ngmutL*-deficient isolates (**B**), respectively. Variable positions are shown by yellow.

**Figure 3 ijms-24-06167-f003:**
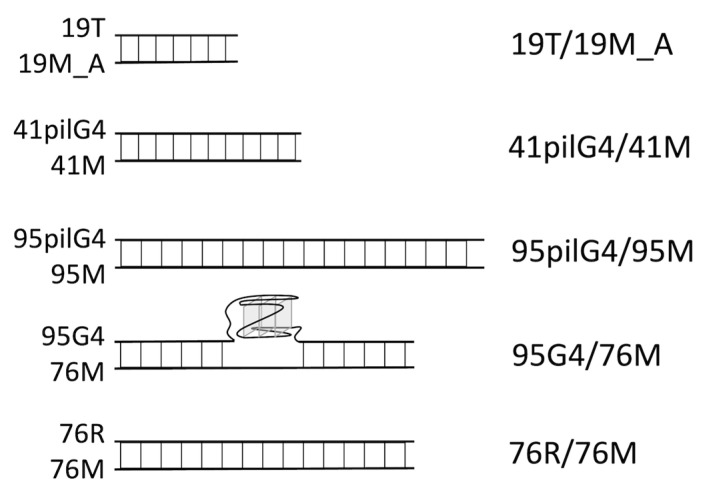
Schematic representations and abbreviations of DNA duplexes (right) formed by hybridization of oligonucleotides (left). The oligonucleotide sequences are shown in Table 1.

**Figure 4 ijms-24-06167-f004:**
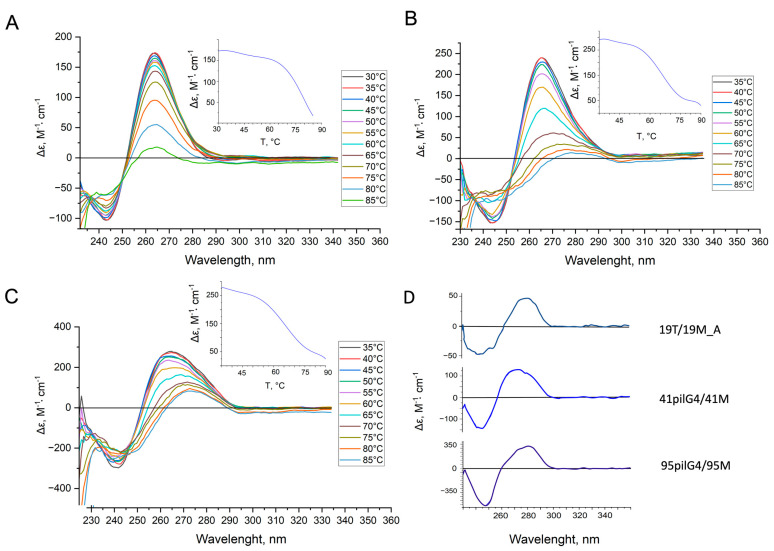
CD spectra of 19pilG4 (**A**), 41pilG4 (**B**), and 95pilG4 (**C**) oligonucleotides recorded at different temperatures in 20 mM HEPES buffer (pH 7.3) containing 5 mM KCl and 140 mM NaCl (~2 µM oligonucleotide strand concentration). Temperature increased from 30 to 85 °C in 5 °C increments; multicolor lines show the CD spectra at different temperatures (**Inserts**). CD-monitored melting profiles at 265 nm. CD spectra of double-stranded DNA models (**D**) (~2 µM oligonucleotide strand concentration) recorded at 37 °C in 20 mM HEPES buffer (pH 8.0) containing 100 mM KCl. It was this buffer that was subsequently used to analyze the interaction of ngMutL with DNAs containing *pilE* G4.

**Figure 5 ijms-24-06167-f005:**
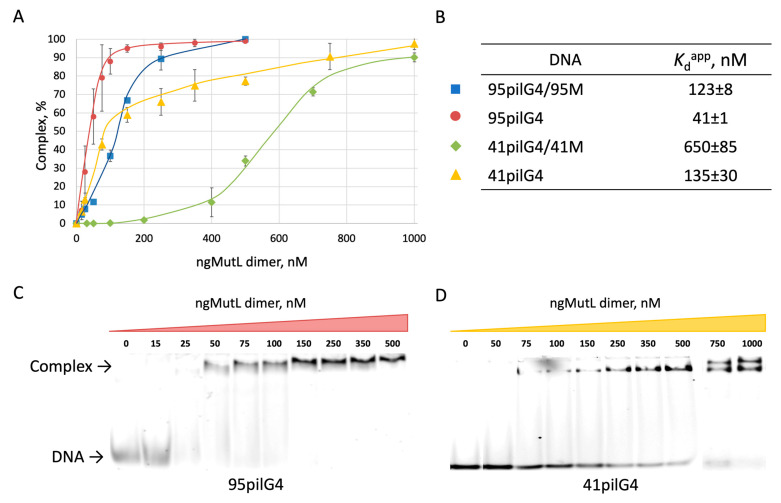
Binding of ngMutL to DNA ligands. The yield of nucleic acid–ngMutL complexes calculated from EMSA data (**A**). TAMRA-labeled DNA probes (20 nM) were incubated for 10 min on ice with 15–1000 nM ngMutL (per dimer) in 20 mM HEPES buffer (pH 8.0) containing 100 mM KCl, 0.5 mg/mL BSA, and 1 mM DTT. Reaction mixtures were analyzed by electrophoresis in 6% polyacrylamide gel in TAE buffer at 4 °C. Error bars represent 95% confidence intervals. The calculation was carried out for at least three independent experiments. *K*_d_^app^ values of ngMutL complexes with single- and double-stranded DNA ligands (**B**). Typical electropherograms of ngMutL binding to 95pilG4 (**C**) and 41pilG4 (**D**).

**Figure 6 ijms-24-06167-f006:**
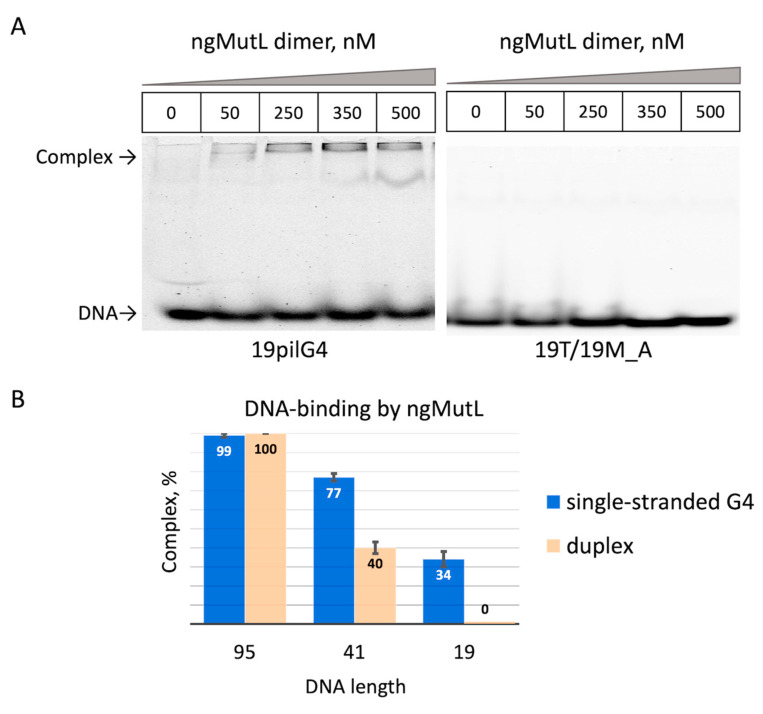
Comparative binding efficiency of ngMutL to single-stranded G4-containing DNA ligands and their duplex versions. Representative electropherograms of ngMutL binding to 19pilG4 and to double-stranded 19T/19M_A (**A**). Efficiency of ngMutL complex formation with 95-, 41-, and 19-mer DNA ligands; both single- and double-stranded (**B**). DNA probes labeled in TAMRA (20 nM) were incubated for 10 min on ice with 500 nM ngMutL (per dimer) in 20 mM HEPES buffer (pH 8.0) containing 100 mM KCl, 0.5 mg/mL BSA, and 1 mM DTT and then analyzed in a non-denaturing 6% polyacrylamide gel in TAE buffer at 4 °C. Error bars represent standard deviations. The calculation was carried out for at least three independent experiments.

**Figure 7 ijms-24-06167-f007:**
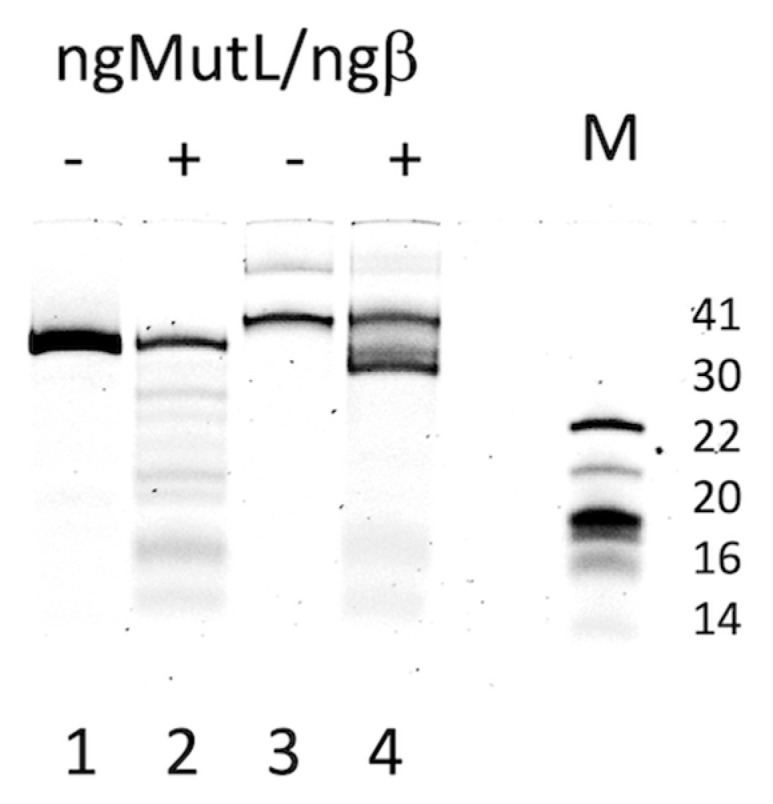
The ngMutL-induced hydrolysis of the DNA substrates (10 nM). The reaction mixtures were incubated in the presence of 0.8 mM ATP, 5 mM MgCl_2_, and 5 mM MnCl_2_ for 90 min at 37 °C, and then analyzed in a 12% polyacrylamide gel containing 7 M urea; 0.25 μM ngMutL (per dimer) was used in the presence of an equimolar amount of ngβ. Electropherograms of 3′-TAMRA-labeled DNA cleavage products of 76R/76M (lanes 1–2) and 95G4/76M (lanes 3–4) are shown. The lengths of DNA markers, M (in nucleotide residues), are indicated on the right.

**Figure 8 ijms-24-06167-f008:**
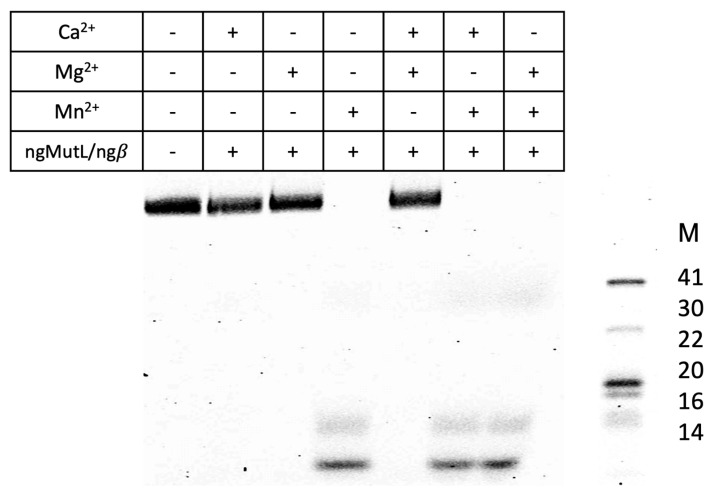
Effect of metal ions Mg^2+^, Mn^2+^, Ca^2+^ and its combinations on ngMutL-induced hydrolysis of 76R/76M (10 nM). The reaction mixtures were incubated in the presence of 0.8 mM ATP, 5 mM metal ions for 90 min at 37 °C, and then analyzed in a 12% polyacrylamide gel containing 7 M urea; 0.5 μM ngMutL (per dimer) was used in the presence of an equimolar amount of ngβ dimer. The lengths of DNA markers, M (in nucleotide residues), are shown.

**Figure 9 ijms-24-06167-f009:**
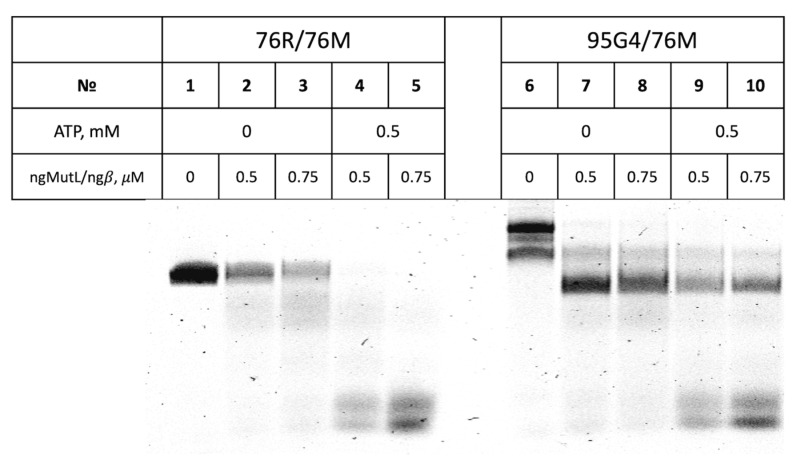
Effect of ATP on ngMutL-induced hydrolysis of the 76R/76M (lanes 1–5) and 95G4/76M (lanes 6–10) (10 nM). The reaction mixtures were incubated in the presence of 0 or 0.5 mM ATP, 5 mM MgCl_2_, and 5 mM MnCl_2_ for 90 min at 37 °C and then analyzed in a 12% polyacrylamide gel containing 7 M urea. Additionally, 0.5 μM ngMutL (per dimer) was used in the presence of an equimolar amount of ngβ dimer. Electropherogram of 3′-TAMRA-labeled DNA cleavage products is shown.

**Table 1 ijms-24-06167-t001:** Abbreviation and sequence of the synthetic DNA oligonucleotides used in this study (G4 motifs highlighted in blue).

Oligonucleotide	Sequence (5′-3′)
19pilG4	AGGGTGGGTTGGGTGGGGA-TAMRA *
19T **	AGTGTGTGTTGTGTGTGGA-TAMRA
19M_A	TCCACACACAACACACACT
41pilG4	ACGCGTTAGAATAGGGTGGGTTGGGTGGGGAATTTTCTATT-TAMRA
41M	AATAGAAAATTCCCCACCCAACCCACCCTATTCTAACGCGT
76R	ATAGGACGCTGACACTGGTGCTTGGCAGCTGAGCCATATGCTCGAGTAACGCTCATAGGATCCAAGCGCGAAAGGA-TAMRA
76M	TCCTTTCGCGCTTGGATCCTATGAGCGTTACTCGAGCATATGGCTCAGCTGCCAAGCACCAGTGTCAGCGTCCTAT
95G4 ***	ATAGGACGCTGACACTGGTGCTTGGCAGCTGAGCCATATTTGGGTGGGTGGGTGGGTTGCTCGAGTAACGCTCATAGGATCCAAGCGCGAAAGGA-TAMRA
95pilG4	GTCGGAATTTGAGATTTTTGAATTTACGCGTTAGAATAGGGTGGGTTGGGTGGGGAATTTTCTATTTTTTAAAAAGCTCCGTTTTCTTGGAAAGC-TAMRA
95M	GCTTTCCAAGAAAACGGAGCTTTTTAAAAAATAGAAAATTCCCCACCCAACCCACCCTATTCTAACGCGTAAATTCAAAAATCTCAAATTCCGAC

* TAMRA fluorophores are shown in pink. ** The 19T differs from 19pilG4 by four G→T substitutions (underlined) in the G4 motif that prevent quadruplex formation. *** The 95G4 contains a d(GGGT)_4_ motif that folds into a parallel quadruplex structure stabilized in a duplex context after hybridization with the partially complementary 76M [24].

## Data Availability

Data is contained within the article or Appendix A.

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
