# Peer review of "pilE G-Quadruplex Is Recognized and Preferentially Bound but Not Processed by the MutL Endonuclease from Neisseria gonorrhoeae Mismatch Repair Pathway"

_ijms, 2023, doi:10.3390/ijms24076167_

Round 1

Reviewer 1 Report

The paper “pilE G-quadruplex-mediated inhibition of Neisseria gonorrhoeae mismatch repair” by Savitskaya et al., reports a study of the interaction of N. gonorrhoeae MutL protein with the quadruplex formed in the promoter area of the pilE gene, an important determinant of immune evasion of this pathogen. The authors show that NgMutL binds but does not cleave the G-quadruplex, and that a lack of MutL increases the genetic variability in the pilE locus. The manuscript is well-written and mostly convincing, but several rather minor issues should be addressed before publishing.

1. The title of the paper is somewhat misleading since the authors do not show “inhibition of mismatch repair”. What is shown is suppression of DNA cleavage by MutL/β-clamp by a quadruplex, and some association between the presence/absence of MutL and pilE variability, which is far from supporting inhibition of mismatch repair in vivo. I recommend changing the title to something more descriptive, such as “pilE G-quadruplex is recognized but not processed by Neisseria gonorrhoeae MutL mismatch endonuclease”.

2. P. 1, lines 33-34: “gonococcus family” is a misnomer; N. gonorrhoeae belongs to the family Neisseriaceae. N. gonorrhoeae are indeed habitually called gonococci but taxonomically no such clade exists.
3. P. 1, line 42: insert “an” between “is” and “important”.
4. P. 4, lines 154-156: delete the fragment of the paper template (“This section…”) left in the text.
5. P. 4, lines 186-187, and p. 15, lines 567-568: there are many ways to numerically represent positionwise amino acid conservation, so please either briefly describe which one is used by Jalview, or provide a reference (preferably _not_ to the Jalview paper but to the original).
6. Fig. 2 and p. 5, line 198: red highlighting is not easily seen in the Figure; I suggest to mark the positions of conserved Cys’s with sticks or some other 3D amino acid representation.
7. P. 5, line 218: replace “in a same” with “in the same”.
8. P. 8, line 287: replace “electropherograms” with “electrophoregram” or, better, “gel image”.
9. Fig. 5A: Please change colors, enlarge symbols, or use different symbol shapes for different data sets. As of now, data for 95pilG4/95M and 41pilG4 are hardly distinguishable.
10. Fig. 5D: What is the origin of two bands at high MutL concentrations? Please discuss in the text.
11. P. 11, line 382, and Fig. 8: There is no sign of activity in the gel with Mg2+ alone (lane 3). Either another gel should be shown, or the claim “Mg2+ also activates the ngMutL endonuclease function” is unsupported.

Reviewer 2 Report

This paper presents an investigation into the role of a G-quadruplex (G4) structure in regulating the antigenic variation of the surface protein PilE in Neisseria gonorrhoeae. The authors use synthetic pilE G4-containing oligonucleotides to study the effect of the G4 structure on ngMutL-mediated regulation of pilin antigenic variation. The study suggests that ngMutL prefers binding to pilE G4 over DNA duplex and that the G4 structure inhibits ngMutL-induced DNA nicking and modulates cleavage positions. While the research hypothesis is sound and the CD and endonuclease activity assays are well performed, there are some concerns that need to be addressed before publication.

Major comments:

1.       The authors analyzed the genomes of N. gonorrhoeae isolates and found that loss of mutS or mutL genes leads to an increase of mutation frequency in pilE, which thus increased the pilE variability. However, the increase of variable substitutions seems very mild (Figure 1B, from 25% to 28% or 36%), and it lacks statistical significance analysis. It would be helpful to combine the conservative and variable substitutions together to show the individual values in each group and perform t-test to compare.

2.       It is unclear why the authors chose the d(GGGT)4 motif for the hydrolysis analysis rather than the pilE G4 motif. Additionally, while the authors claim that the DNA:ngMutL ratio of 1:25 provides 100% endonuclease activity on 76R/76M substrate, the data in Figure 7 does not show 100% hydrolysis activity even though the DNA:ngMutL ratio is 1:25 as indicated in the figure legend.

Minor comments:

1.       Line 492, typo, “EMCA” should be “EMSA”

2.       Specify the “N” value in the figures with error bars. (Figure 5A, Figure 6B)

Round 2

Reviewer 2 Report

No more comments